# Nanoscale architecture and coordination of actin cores within the sealing zone of human osteoclasts

**Marion Portes[1], Thomas Mangeat[2], Natacha Escallier[1], Ophélie Dufrancais[1], Brigitte Raynaud-Messina[1], Christophe Thibault[3], Isabelle Maridonneau-Parini[1], Christel Vérollet[1]\*, Renaud Poincloux[1]\***

[1]Institut de Pharmacologie et de Biologie Structurale, Université de Toulouse, CNRS, UPS, Toulouse, France; [2]LITC Core Facility, Centre de Biologie Intégrative, Université de Toulouse, CNRS, UPS, Toulouse, France; [3]LAAS-CNRS, Université de Toulouse, CNRS, INSA, Toulouse, France

**Abstract** Osteoclasts are unique in their capacity to degrade bone tissue. To achieve this process, osteoclasts form a specific structure called the sealing zone, which creates a close contact with bone and confines the release of protons and hydrolases for bone degradation. The sealing zone is composed of actin structures called podosomes nested in a dense actin network. The organization of these actin structures inside the sealing zone at the nano scale is still unknown. Here, we combine cutting-edge microscopy methods to reveal the nanoscale architecture and dynamics of the sealing zone formed by human osteoclasts on bone surface. Random illumination microscopy allowed the identification and live imaging of densely packed actin cores within the sealing zone. A cross-correlation analysis of the fluctuations of actin content at these cores indicates that they are locally synchronized. Further examination shows that the sealing zone is composed of groups of synchronized cores linked by α-actinin1 positive filaments, and encircled by adhesion complexes. Thus, we propose that the confinement of bone degradation mediators is achieved through the coordination of islets of actin cores and not by the global coordination of all podosomal subunits forming the sealing zone.

**\*For correspondence:**
verollet@ipbs.fr (CV);
poincloux@ipbs.fr (RP)

**Competing interest:** The authors declare that no competing interests exist.

## Editor's evaluation

The authors present here an elegant study in which they analyze the structure of human osteoclast podosomes at the nanoscale, combining microscopy methods to reveal the architecture and dynamics of the sealing zone on the bone surface. Through random illumination microscopy, the live imaging of densely packed actin cores within the sealing zone shows that these cores are locally synchronized, connected by α-actinin filaments, and surrounded by adhesion complexes. The authors propose a model in which the function of podosomes during bone resorption is accomplished through the coordination of islets of the actin core and not through the global coordination of all podosome subunits that form the sealing zone. This article has the potential to generate a significant impact in the field of osteoclast biology.

## Introduction

Osteoclasts are giant multinucleated cells of the hematopoietic lineage, specialized in the degradation of bone matrix. To do so, protons accumulate into the resorption lacuna thanks to the vacuolar H[+]-ATPase (v-ATPase), thus lowering the local pH and facilitating the solubilization of apatite, the main

mineral component of bone. This acidic environment is also prone to enhance the digestion of bone organic matrix by proteases secreted by osteoclasts such as cathepsin K (*Teitelbaum, 2007*; *Soysa and Alles, 2016*). The efficiency of this process relies on the ability of the cell to create an enclosed resorption compartment, *via* the formation of a unique cytoskeletal structure, the sealing zone (*Jurdic et al., 2006*; *Delaisse et al., 2021*).

First described as a subcellular entity made of electron dense material and apparently deprived of any organelle, thus resulting in first denomination as the 'clear zone', the sealing zone was then revealed to consist in a dense accumulation of actin filaments forming a circular shape surrounding the resorption lacuna (*Zambonin-Zallone et al., 1988*; *Kanehisa et al., 1990*; *Teti et al., 1991*; *King and Holtrop, 1975*; *Han et al., 2019*). Examination with scanning electron microscopy (SEM) of cells removed of their basal membrane brought to light the peculiar arrangement of actin filaments within this structure, and particularly unveiled the existence of a dense network of podosomes composing this structure (*Luxenburg et al., 2007*; *Akisaka and Yoshida, 2015*; *Akisaka and Yoshida, 2019*). Podosomes are adhesion structures that are generally scattered in macrophages and dendritic cells. Podosome typical components, such as vinculin, paxillin, talin, or cortactin, are also localized in the sealing zone (*Chabadel et al., 2007*; *Saltel et al., 2008*; *Lakkakorpi et al., 1993*; *Lakkakorpi et al., 2001*; *Pfaff and Jurdic, 2001*; *Hurst et al., 2003*; *Gil-Henn et al., 2007*; *Ory et al., 2008*; *Ma et al., 2010*). In a distinct way compared to single podosomes, vinculin, paxillin, and talin were reported to form a 'double circle' flanking the sealing zone on each side, while cortactin mainly colocalized with actin in between (*Jurdic et al., 2006*; *Lakkakorpi et al., 1993*; *Pfaff and Jurdic, 2001*; *Lakkakorpi et al., 1999*; *Lakkakorpi and Väänänen, 1996*; *Tehrani et al., 2006*). Noteworthy, most of the studies on the bone degradation machinery were carried out based on observations of osteoclasts on glass substrates instead of bone, due to the lack of optical transparency and high auto-fluorescence of the mineralized matrix. As a result, only poor knowledge has been collected about the architecture and dynamics of the functional sealing zone. Indeed, it has been suggested that actin structures on bone differ from the ones formed on glass, mainly in their total width, and the interconnectivity and density of actin cores within (*Luxenburg et al., 2007*). Therefore, only structures formed on bone or bone-mimicking materials are called sealing zone, structures on glass being denominated as 'sealing zone like' or podosome belts (*Jurdic et al., 2006*; *Saltel et al., 2004*).

How podosomes are organized and coordinated to allow efficient sealing and bone degradation is therefore still unknown. Hence, it appears paramount to develop higher resolution microscopy techniques compatible with observation on bone substrates. This could yield valuable information concerning the spatial distribution of major actin-binding proteins within the sealing zone, otherwise only arduously accessible *via* electron microscopy and correlative microscopy *Luxenburg et al., 2007*; *Akisaka and Yoshida, 2015*; *Akisaka and Yoshida, 2019*; *Akisaka and Yoshida, 2016*; *Geblinger et al., 2009*. Additionally, observation of the sealing zone internal dynamics would provide substantial hints to understand the sealing ability of such a structure. This exploration would require both a spatial and temporal high-resolution microscopy technique.

In this work, we used cutting-edge super-resolution microscopy methods to reveal the architecture and dynamics of the bone degradation machinery formed by human osteoclasts. First, we examined the three-dimensional (3D) nanoscale organization within the podosome belt thanks to a single-molecule localization method. Then, random illumination microscopy (RIM) acquisitions of human osteoclasts plated on bone allowed to resolve single actin cores composing the sealing zone. RIM technique also proved to be efficient in deciphering this nanoscale organization in living samples. Hence, cross-correlation analysis of the fluctuation of the sealing zone actin content could show that cores were locally synchronized. Further analysis of the organization of adhesion components and actin crosslinkers revealed that the sealing zone is composed of coordinated groups formed by α-actinin1-linked podosomal cores and encircled by adhesion complexes. Therefore, confinement of bone degradation enzymes may be achieved through the alternating contact of functional islets of actin cores.

## Results

### Three-dimensional nanoscale organization of the podosome belt in human osteoclasts

The sealing zone formed by bone-degrading osteoclasts is described as a dense network of actin cores that appear fused under conventional microscopy, surrounded by two lines of adhesion sites (*Jurdic et al., 2006*; *Lakkakorpi et al., 1993*; *Pfaff and Jurdic, 2001*; *Lakkakorpi et al., 1999*; *Lakkakorpi and Väänänen, 1996*; *Tehrani et al., 2006*). The precise architectural and dynamics of this structure are almost uncharacterized. To address the organization of osteoclast podosomes, a 3D super-resolution method called DONALD was used. DONALD is a single molecule localization method combining direct stochastic optical reconstruction microscopy for the in-plane detection of proteins, and SAF analysis to gain access to the absolute axial position of fluorophores relative to the glass coverslip. This nanoscopy technique thus benefits from approximately 15 nm localization precision in the three dimensions (*Bourg et al., 2015*). Human osteoclasts derived from blood monocytes were differentiated for 10 days, plated on glass and observed with this 3D super-resolution technique. On glass, osteoclasts form structures called podosome belts, sharing certain characteristics with sealing zones (*Jurdic et al., 2006*). In particular, podosome belts also exhibit areas where podosome cores appear fused, that is when sets of several cores are brought together inside a single and large ring of adhesion sites, similar to the described structure of the sealing zone. Indeed, this fusion of podosome cores was observed in 9.4+/-3.5% of multinucleated human osteoclasts. We therefore focused on the zones displaying podosome fusion (*Figure 1A–D*).

Spatial distributions of key structural components of podosomes, namely cortactin, α-actinin1, filamin A, vinculin, paxillin and the C-terminal extremity of talin (talin-C), whose height was previously correlated with podosome protrusion forces (*Bouissou et al., 2017*), were thus explored using DONALD. Image analysis consisted in localizing the various target proteins with respect to the actin cores composing the podosome belts (*Figure 1—figure supplement 1*). Cortactin in-plane distribution was characterized by an accumulation in the first 250 nm surrounding the actin cores, and kept at a rather constant height of approximately 164 nm while actin height peaked at the core and then decreased by approximately 50 nm on both sides (*Figure 1—figure supplement 2A-D*). α-actinin1 was preferentially localized in the vicinity of the actin cores, up to 500 nm in distance. Similarly to cortactin, its height stayed approximately constant throughout the distance profile, at approximately 126 nm (*Figure 1—figure supplement 2E-F*).

Vinculin, paxillin, talin, and filamin A were mostly absent from the regions of dense actin staining, and encircled multiple actin cores (*Figure 1A–G*, *Figure 1—figure supplement 2G-J*). Vinculin and talin-C heights declined by nearly 20 nm toward the inner part of the cell (*Figure 1E and F*). In contrast, filamin A and paxillin, for which average heights were 45 nm and 139 nm, respectively, exhibited almost no change in height when away from the belt center (*Figure 1H–I*, *Figure 1—figure supplement 2G-J*). Of note, our analyses revealed little difference in terms of organization between the internal and external distributions of the different podosome components analyzed, the most striking being a closer proximity of the vinculin to the cores on the outer side of the belt, suggesting that the podosome belt is overall a symmetric structure.

### Nanoscale organization of actin cores in the sealing zone

To further explore the architecture of the sealing zone, we then investigated the organization of podosomes in mature human osteoclasts plated on bovine bone slices. After 3 days, osteoclasts efficiently degraded bone, as shown by scanning electron microscopy (SEM) observations (*Figure 2A* and *Figure 2—figure supplement 1*). SEM acquisitions of unroofed cells confirmed that the sealing zones formed by human osteoclasts are composed of individual F-actin cores (*Figure 2B*), similarly to what was shown in osteoclasts differentiated from the mouse cell line RAW 264.7 or harvested from rabbit long bones (*Luxenburg et al., 2007*; *Akisaka and Yoshida, 2019*). These cores were nested in a dense network of actin filaments, and appeared connected to their neighbors by filaments running parallel to the substrate (*Figure 2B'*, arrowheads). Morphometric characterization of the podosome network involves defining the podosome density, the position of each podosome core, and their position relative to one another. Thus, these characteristics were assessed by manually encircling each core and estimating their radius from the selected area. Analysis of 457 cores in nine different cells yielded a median core radius of 114 nm (*Figure 2D*), that is two times smaller than the radius of macrophage

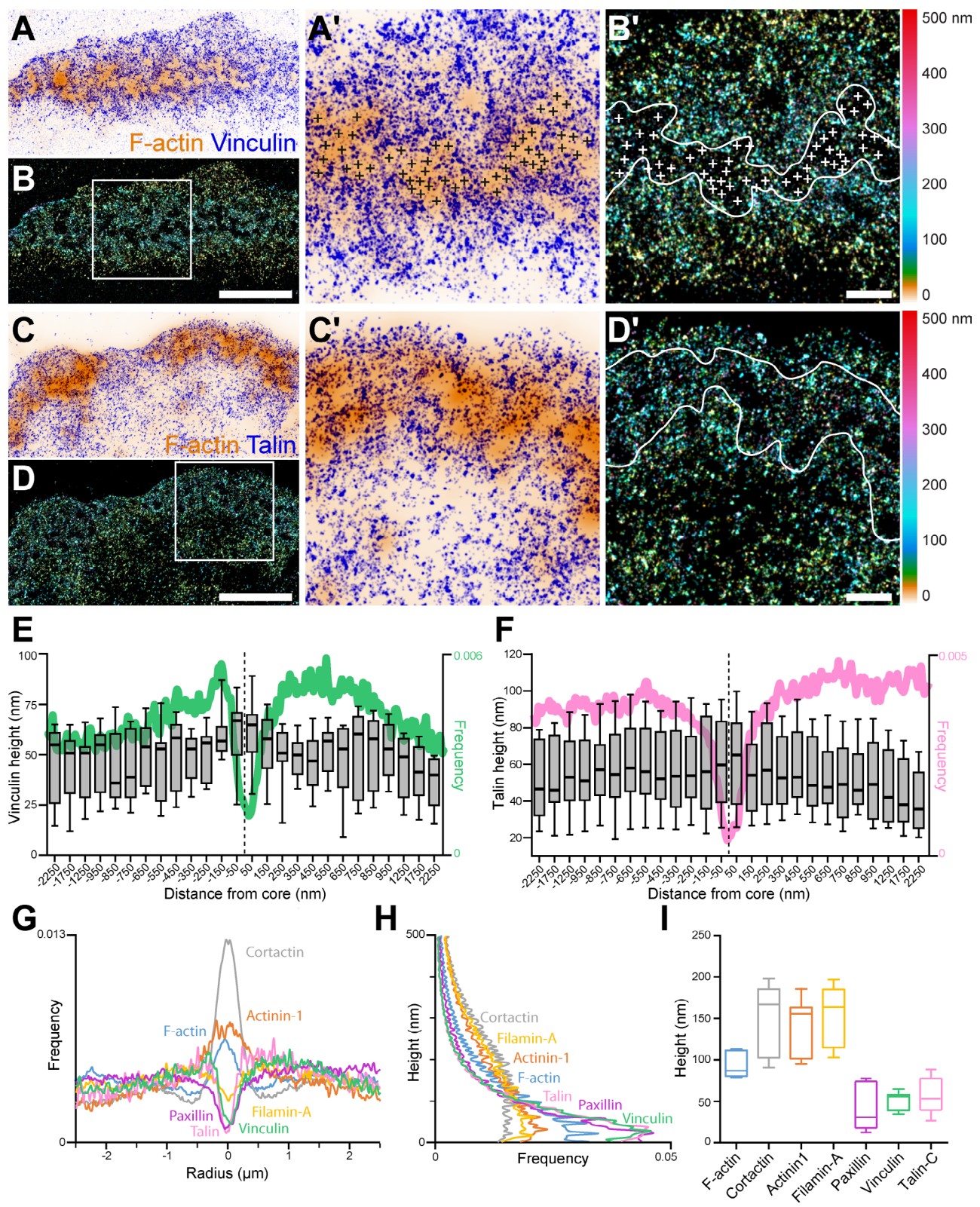

**Figure 1.** 3D nanoscopy of vinculin and talin-C in the osteoclast podosome belt. (**A**) Representative dSTORM images of vinculin (blue) merged with the corresponding epifluorescence images of the F-actin cores (ochre). (**A′**) Enlarged view of (**A**). The white crosses indicate the localization of actin cores. (**B**) DONALD images corresponding to (**A**) where the height is represented in false color (scale shown in **B′**). (**B′**) Enlarged view of (**B**). The white crosses indicate the localization of actin cores. (**C**) Representative dSTORM images of talin-C (blue) merged with the corresponding epifluorescence images of

*Figure 1 continued on next page*

*Figure 1 continued*

the F-actin cores (ochre). (**C'**) Enlarged view of (**C**). (**D**) DONALD images corresponding to (**A**) where the height is represented in false color (scale shown in (**D'**)). (**D'**) Enlarged view of (**D**). (**E–F**) Height profiles for vinculin (**E**) and talin-C (**F**) and radial distributions (in green and pink, respectively) with respect to the distance to the center of the podosome belt. Data from 465 cores and 1235 cores were quantified for vinculin and talin-C graphs, respectively. (**G–H**) Radial (**G**) and vertical (**H**) distributions of cortactin, α-actinin1, filamin A, paxillin, vinculin, and talin-C in podosome belts. (**I**) Median axial positions of F-actin, cortactin, α-actinin1, filamin A, paxillin, vinculin, and talin-C in podosome belts. Box-and-whisker plots show the median, lower and upper quartiles (box) and the 10th and 90th percentiles (whiskers). Scale bars: 5 µm (**A–D**), 1 µm (**B', D'**).

The online version of this article includes the following source data and figure supplement(s) for figure 1:

**Source data 1.** *Figure 1* source data for all protein height profiles.

**Source data 2.** *Figure 1* - source data for all protein radial and vertical distributions.

**Figure supplement 1.** Analysis workflow for the 3D localization of proteins relative to actin cores with DONALD.

**Figure supplement 2.** 3D nanoscopy of F-actin, cortactin, α-actinin1, filamin A, and paxillin in the osteoclast podosome belt.

podosomes (*Proag et al., 2015*; *Jasnin et al., 2021*). Furthermore, applying Delaunay's tessellation to each dataset allowed for the characterization of the average inter-core distances within the same cell (*Figure 2B' and C"*). Direct neighbor pairs were 705 nm apart (*Figure 2E*), and first neighbors were 443 nm apart (*Figure 2F*), similarly to what was described previously (*Deguchi et al., 2019*).

F-actin distribution in human osteoclasts unroofed on bone was then observed by a new super-resolution method called random illumination microscopy (RIM, *Figure 2C* and *Figure 2—figure supplement 2*). RIM consists in illuminating a sample with a series of random speckles and processing the stack of images using signal processing and statistical tools, to gather a lateral and axial resolution of 100 and 300 nm, respectively. This method benefits from similar resolution as with traditional structured illumination microscopy (SIM), while not requiring any initial calibration step. In addition, its super-resolution capability allows for characterization of events within thick samples, until 30 µm in unknown optical medium (*Mangeat et al., 2021*). RIM thus enabled super-resolution imaging of the functional sealing zone. Actin staining showed a dense though discontinuous thick pattern within the sealing zone (*Figure 2C*, *Figure 2—figure supplement 2A-B*, *Video 1*), in which single actin structures could be spotted. Localization of lysosomal membrane protein LAMP1 in bone-resorbing osteoclasts confirmed the accumulation of LAMP1 compartments in the middle of the sealing zone (*Figure 2—figure supplement 2*). Using this approach, the number of podosomes, the average diameter of their core were determined, and from these parameters, their relative spatial distribution was estimated. More specifically, signal analysis of the actin staining localized local intensity maxima, the geometric features of which were assessed by extracting both the coordinates of the local maxima (*Figure 2C'*), and the signal intensity values along eight directions, evenly distributed from the center. Signal variations were quantified along these 1 µm long segments, the half-width of the peak in each direction was computed after spatial derivation, and the peak radii were computed by averaging the eight values (*Figure 2—figure supplement 3*). The radius distribution yielded an average value of 97 nm (*Figure 2D*). Delaunay's tessellation was applied to the peak coordinates to characterize their spatial arrangement (*Figure 2C"*). Direct neighbors were in a 694 nm distance range (*Figure 2E*), with first neighbors being 399 nm apart (*Figure 2F*).

As the distribution of radii and neighbor distances were similar with SEM and RIM, we concluded that RIM efficiently allows for the observation of single actin cores within the sealing zones of human osteoclasts.

## Local synchrony of sealing zone actin cores

We then used RIM to image living human osteoclasts adhering on bone to address the dynamics of actin cores within the sealing zone. Human osteoclasts transduced with GFP-tagged LifeAct lentiviruses and actively degrading bone slices were first observed by widefield fluorescence microscopy over 30 min. Subsequent deconvolution of the images and color-coding for time using a rainbow scale revealed that cores appeared stable over at least 30 min and analysis of kymographs along the sealing zone revealed actin intensity oscillations during the entire duration of the acquisitions (*Figure 3A*, *Video 2*). To further characterize this dynamic process at a higher spatial and temporal resolution, small regions of the sealing zone were observed with RIM. Variations of the actin content in podosome

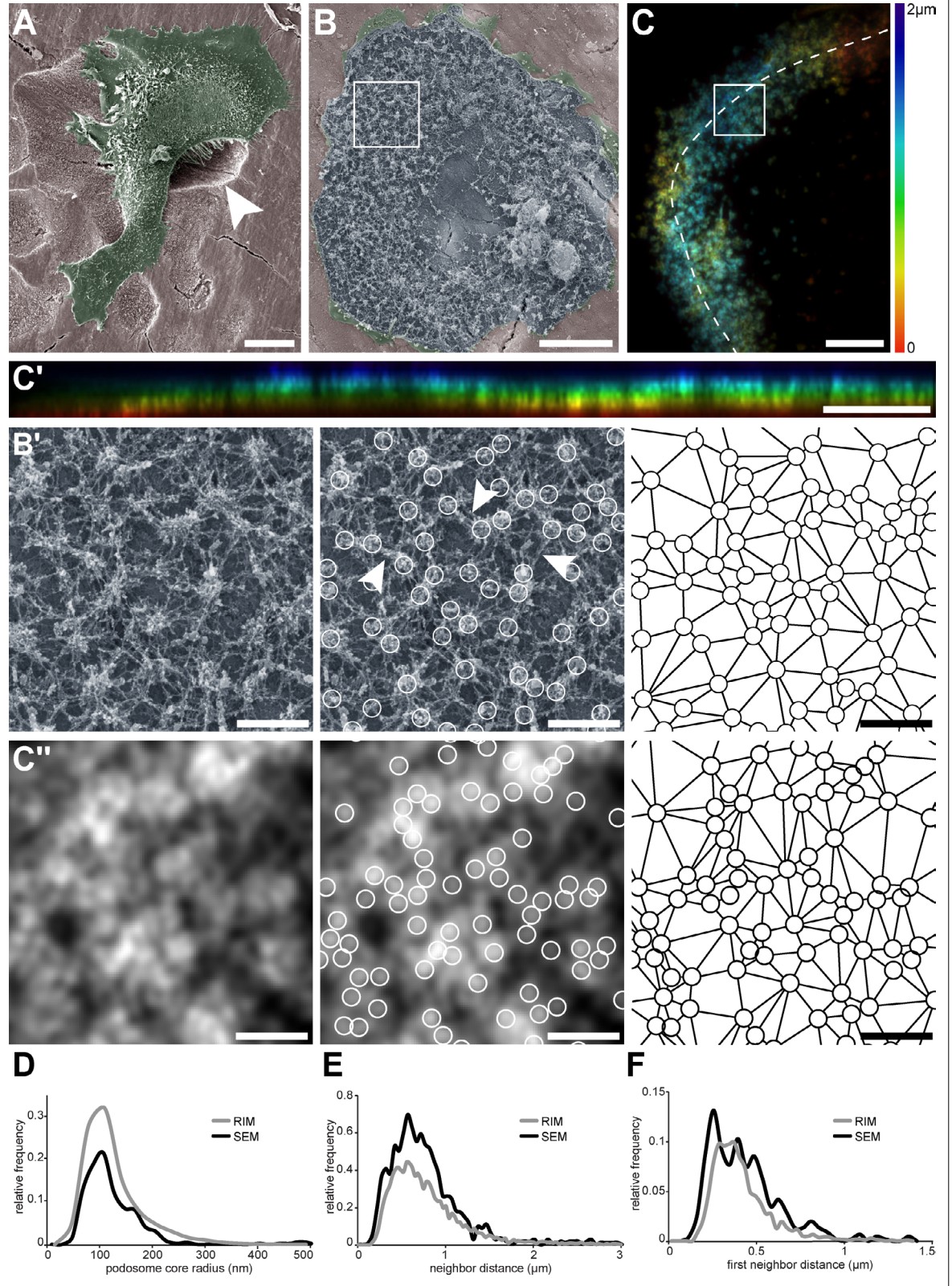

**Figure 2.** Nanoscale organization of actin cores in the sealing zones formed by human osteoclasts. (**A**) Pseudo-colored scanning electron micrograph of a human monocyte-derived osteoclast degrading bone. Arrowhead shows bone degradation. (**B**) Scanning electron microscopy image of an unroofed osteoclast. Of note, the ruffled border could not be preserved by the unroofing procedure. (B', left) Enlarged view of (**B**). (B', middle) Localization of actin cores (circles). Arrowheads point to lateral actin filaments linking actin cores together. (B', right) Delaunay triangulation from the cores.

*Figure 2 continued on next page*

*Figure 2 continued*

(**C**) Projection of a z-stack of an osteoclast stained for F-actin and adhering on bone, acquired every 200 nm and reconstructed by RIM. Color codes for height using a rainbow scale. (**C'**) Orthogonal projection along the line marked in (**C**). The same color code for height was used. (**C''**, left) Enlarged view of (**C**). (**C''**, middle) Localization of actin cores (circles). (**C''**, right) Delaunay triangulation from the cores in (**C''**). (**D**) Histogram of the core radii, as measured by SEM (black) and RIM (grey). (**E**) Histogram of the average distances to direct neighbors measured as Delaunay edges, as measured by SEM (black) and RIM (grey). (**F**) Histogram of the average distances to first neighbors, as measured by SEM (black) and RIM (grey). In (**D–F**), 457 and 2781 cores were quantified for SEM and RIM, respectively. Scale bars: 20 µm (**A**), 5 µm (**B, C, C'**), 1 µm (**B', C''**).

The online version of this article includes the following source data and figure supplement(s) for figure 2:

**Source data 1.** *Figure 2* - source data for SEM and RIM morphometric histograms.

**Figure supplement 1.** Bone degradation by human osteoclasts.

**Figure supplement 2.** Comparison between epi-fluorescence and RIM super-resolution microscopy and localization of LAMP1.

**Figure supplement 3.** Analysis workflow for the geometric characterization of actin cores with RIM technique.

---

cores were measured and appeared to fluctuate synchronously between neighbors (*Figure 3B* and *Video 3*, left).

To quantify to what extent podosome actin content varied concomitantly between neighbors, 2840 cores distributed in 10 different sealing zones were localized, and their associated actin intensity signal was extracted for further analysis. The distance between every possible core pair was estimated and their respective signals were compared using Pearson cross-correlation analysis. A high positive Pearson coefficient corresponds to acute temporal synchrony for the analyzed pair of actin signals. Highest Pearson coefficient values between 1 and 0.37, corresponding to half of the maximum mean value, were obtained for cores within a 700 nm radius distance. They slowly decreased to approximately 0.10 at longer distances (*Figure 3C*). This result hinted at the existence of a spatial synchrony between neighbors within the sealing zone, comparable to observations for podosomes in human macrophages (*Proag et al., 2015*). Moreover, potential periodicity of actin content fluctuations was examined through analysis of the Fourier spectra corresponding to live acquisitions. This yielded three specific frequencies that seemed to prevail: 0.01 Hz, 0.04 Hz, and 0.15 Hz, corresponding to periods of approximately 100 s, 25 s, and 7 s, respectively. In macrophages, similar oscillation periods had previously been identified by examining the stiffness variations of single podosomes (*Labernadie et al., 2014*). In order to obtain a graphical representation of podosome synchrony, differential films were assembled by subtracting 2 sequential time points, therefore representing local actin intensity gradients. In *Figure 3D*, orange stands for a positive gradient, that is local polymerization, and blue represents a negative gradient, that is local depolymerization. Strikingly, polymerization and depolymerization regions appeared as clusters with slowly varying areas, containing a few actin cores (*Figure 3D*, *Video 3*, right). Actin polymerization and depolymerization processes thus appeared to be synchronous within actin core clusters.

In conclusion, we showed a local synchrony of F-actin oscillations between cores of the same superstructure. This suggests the organization of actin cores into functional clusters within the sealing zone.

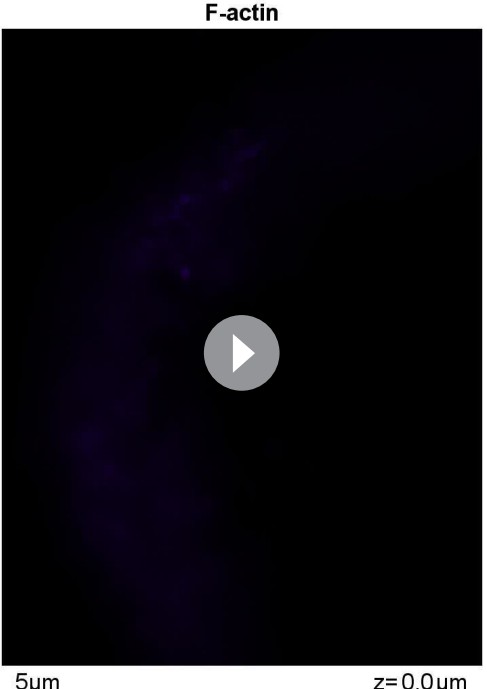

**Video 1.** Z-stack by RIM of a human osteoclast adhering on bone and stained for F-actin. The movie shows a z-stack of an osteoclast stained for F-actin and adhering on bone acquired by RIM microscopy at 200 nm intervals. Each plane of this z-stack has been colored with a different color. Color-coded for height using a rainbow scale.

https://elifesciences.org/articles/75610/figures#video1

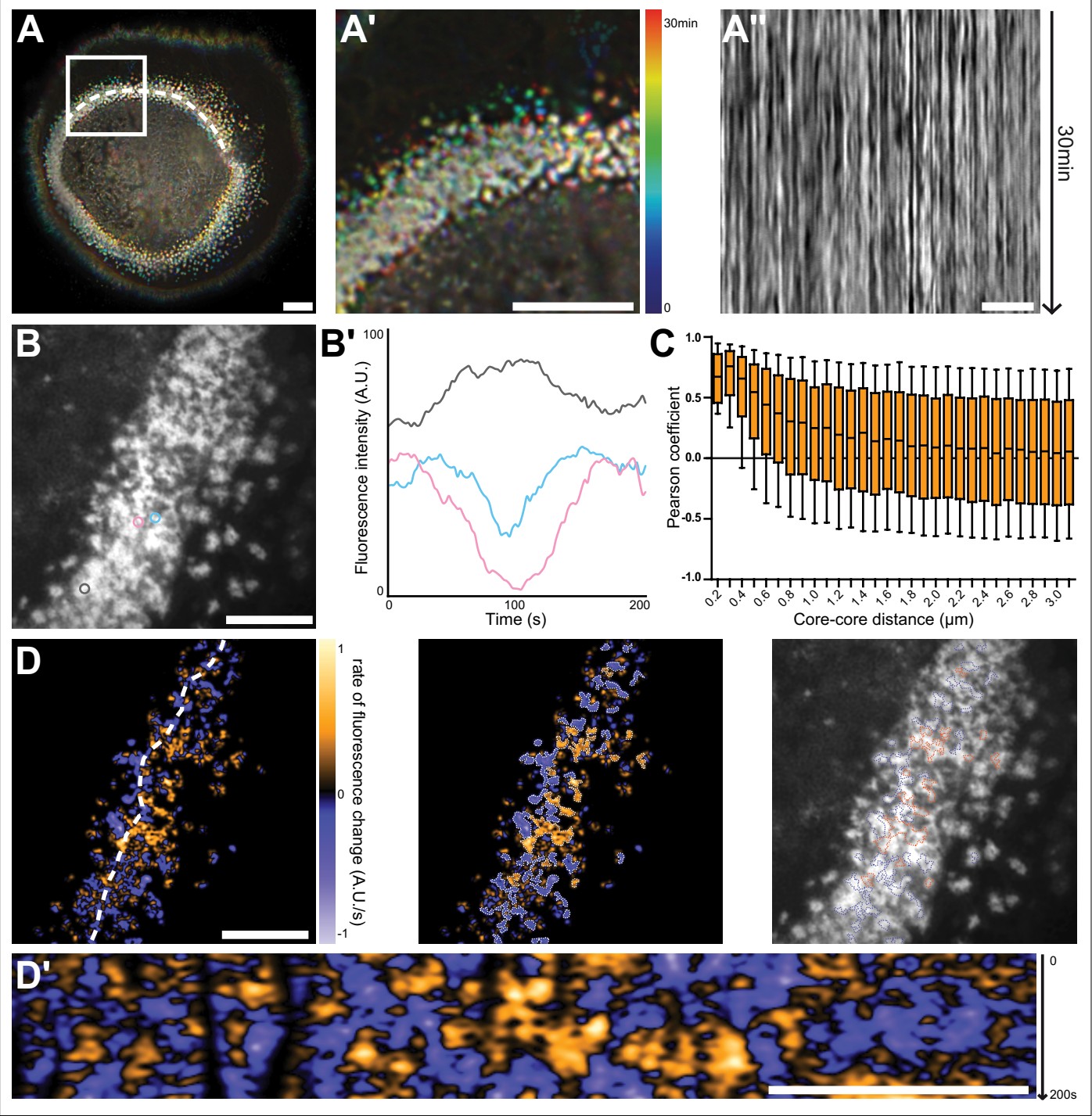

**Figure 3.** Nanoscale analysis of the dynamics of the sealing zone. (**A**) Temporal projection of deconvolution images of a sealing zone acquired over 30 min, color-coded for time using a rainbow scale. Thus, the structures that remain at the same spot tend to appear whiter, whereas short-lived or mobile podosomes remain colored. (**A'**) Enlarged view of (**A**). (**A''**) Kymograph along the line marked in (**A**). (**B**) RIM image of a sealing zone stained for F-actin with LifeAct-GFP. (**B'**) Measurements of LifeAct-GFP intensity variations of the 3 actin cores marked in (**B**). (**C**) Pearson coefficients of actin intensity fluctuations of podosome pairs as a function of distance between pairs. Data for a total of 2839 cores were quantified. (D, left) Image of the rate of fluorescence change corresponding to the cell shown in (**B**). (D, middle) Segmentation of the growing and decreasing clusters of actin cores. (D, right) Superimposition of the RIM image with the segmented regions of coordinated actin clusters shown in (**D''**). (**D'**) Kymograph along the line marked in (**D**). Box-and-whisker plots show the median, lower and upper quartiles (box) and the 10th and 90th percentiles (whiskers). Scale bars: 10 µm (**A**), 5 µm (**A'-D'**).

The online version of this article includes the following source data for figure 3:

*Figure 3 continued on next page*

*Figure 3 continued*

**Source data 1.** *Figure 3* - source data for LifeAct intensity variations of 3 cores.

**Source data 2.** *Figure 3* - source data for LifeAct intensity variations of 3 cores.

## Sealing zone actin cores are organized into islets surrounded by adhesion complexes

Although no partitioning of podosome cores was evident solely on the basis of the actin staining, we then reasoned that, since groups of actin cores display local synchrony, a specific organization of these cores could be revealed by localizing adhesion components of the sealing zone. Spatial distributions of the same components of the sealing zone that were imaged with the DONALD technique (*Figure 1*) were thus explored by RIM, and we developed a quantitative image workflow to analyze the localization of these proteins with respect to the actin cores in human osteoclasts adhering on bone. Protein localizations were evaluated along 1.5 µm long and 100 nm wide lines, in longitudinal and transverse directions relative to the local sealing zone orientation. Because sealing zones were different in terms of actin core density, the actin core width was normalized, allowing for the localization of the target proteins with respect to the core domain (see vinculin, *Figure 4—figure supplement 1*). Cortactin was localized within the core domain, as had been previously reported (*Hurst et al., 2003*), and displayed a wider distribution compared to the average core diameter (*Figure 4A and E* and *Figure 4—figure supplement 2*). α-actinin1 mostly colocalized with actin at the close periphery of the core, and appeared less present in the most central part of the core (*Figure 4B and E* and *Figure 4—figure supplement 2*). Filamin A, vinculin, paxillin and talin were preferentially encircling multiple actin cores, with little staining in between cores in the inner part of the sealing zone (*Figure 4C–D and E*, *Figure 4—figure supplements 2–3*). Measurements of these islets of clustered cores showed that they were 2.3+/-2.1 µm² (average +/-SD) and contained 7+/-8 (average +/-SD) cores. These observations showed a strong divergence with the 'double circle' distribution described before (*Lakkakorpi et al., 1993*; *Lakkakorpi and Väänänen, 1996*), as the sealing zone appeared to be composed of islets of actin cores that are bordered by a network of adhesion complexes.

In scattered podosomes formed by macrophages or dendritic cells, there is a local co-regulation between polymerization of actin in the core and formation of surrounding adhesion sites (*Bouissou et al., 2017*; *van den Dries et al., 2013*). To determine whether such a local co-regulation takes place inside the islets of podosomes in the sealing zone, we first measured local intensities of cortactin, α-actinin1, filamin A, vinculin,

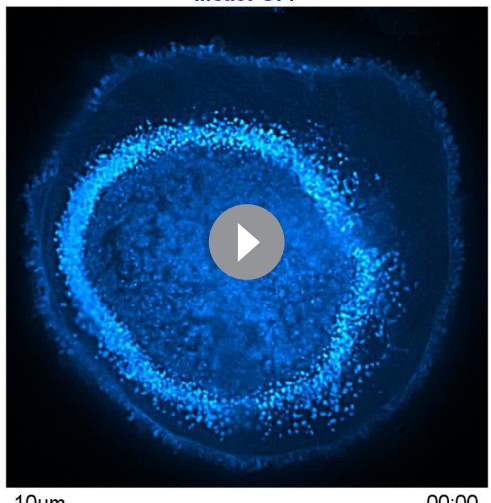

**human osteoclast on bone**
**lifeact-GFP**

10µm                00:00

**Video 2.** Deconvolution time-series of a sealing zone over 30 min. Time-series movie of a human osteoclast expressing LifeAct-GFP and adhering on bone. The video was acquired by wide-field fluorescence microscopy at 2 s intervals during 30 min and deconvoluted.

https://elifesciences.org/articles/75610/figures#video2

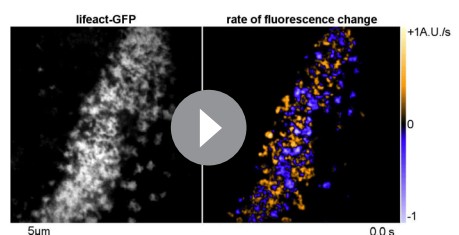

lifeact-GFP        rate of fluorescence change        +1A.U./s

5µm                0.0 s        -1

**Video 3.** RIM time-series and rate of fluorescence change of the F-actin content of a sealing zone. Reconstruction at 2.4 s intervals during 160 s of a time-series acquired by RIM microscopy. Left panel: RIM images of a sealing zone stained for F-actin with LifeAct-GFP. Right panel: images of the rate of fluorescence change. Orange stands for a positive gradient, that is local actin polymerization, and blue represents a negative gradient, that is local actin depolymerization.

https://elifesciences.org/articles/75610/figures#video3

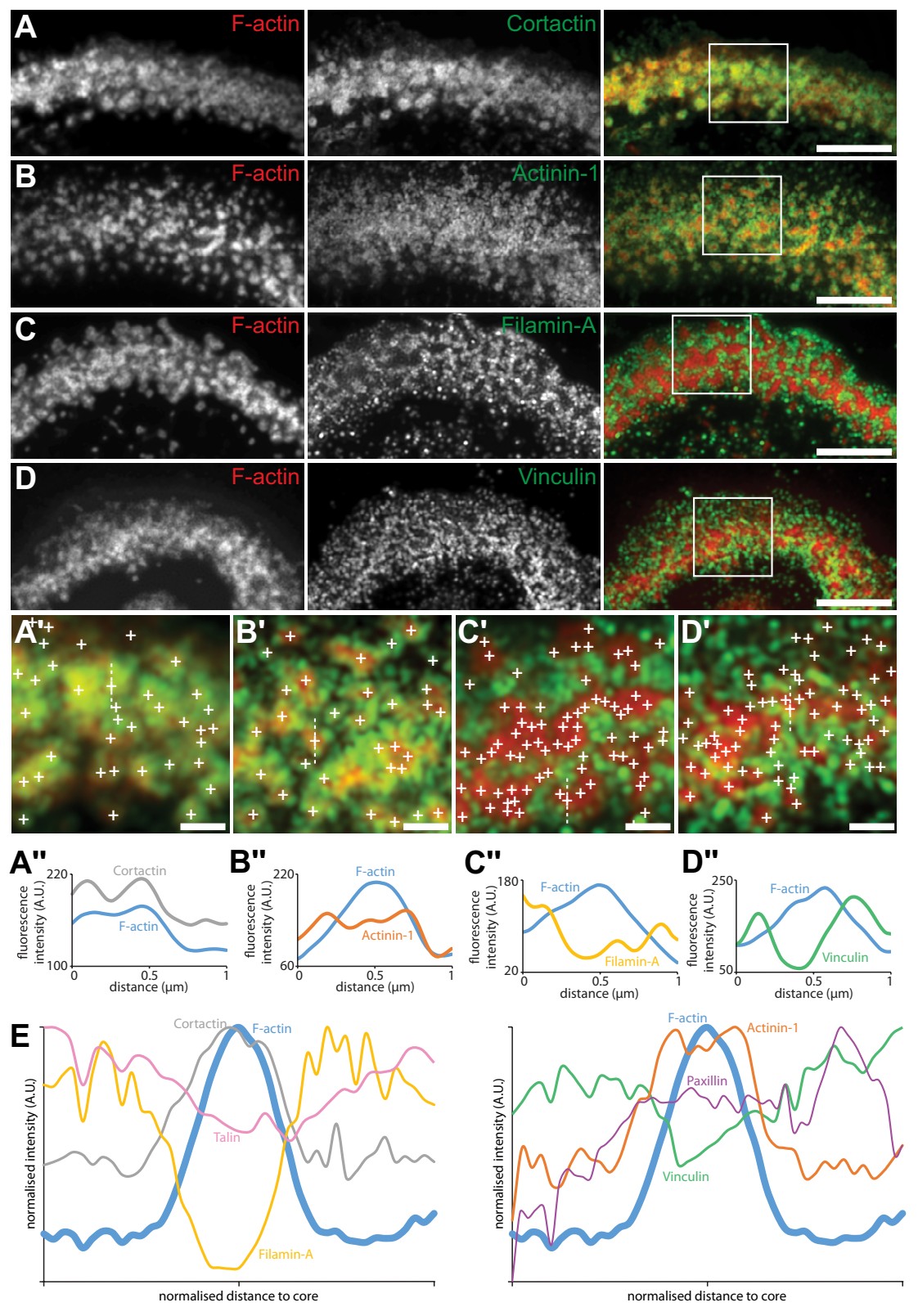

**Figure 4.** Localization in the sealing zone of actin core and ring proteins. (**A**) Representative RIM images of sealing zones co-stained for F-actin (red) and cortactin (green). (**A'**) Enlarged view of (**A**) where white crosses indicate the localization of actin cores. (**A''**) Intensity profiles along the dotted line marked in (**A'**). (**B**) Representative RIM images of sealing zones co-stained for F-actin (red) and α-actinin1 (green). (**B'**) Enlarged view of (**B**) where white crosses indicate the localization of actin cores. (**B''**) Intensity profiles along the dotted line marked in (**B'**). (**C**) Representative RIM images of sealing zones co-

*Figure 4 continued on next page*

*Figure 4 continued*

stained for F-actin (red) and filamin A (green). (**C′**) Enlarged view of (**C**) where white crosses indicate the localization of actin cores. (**C″**) Intensity profiles along the dotted line marked in (**C′**). (**D**) Representative RIM images of sealing zones co-stained for F-actin (red) and vinculin (green). (**D′**) Enlarged view of (**D**) where white crosses indicate the localization of actin cores. (**D″**) Intensity profiles along the dotted line marked in (**D′**). (**E**) Normalized intensity profiles of F-actin, cortactin, α-actinin1, filamin A, vinculin, paxillin, and talin (medians of 1080, 239, 265, 277, 299, 197 and 988 cores for each staining, respectively). Scale bars: 5 µm (**A, B, C, D**), 1 µm (**A′, B′, C′, D′**).

The online version of this article includes the following source data and figure supplement(s) for figure 4:

**Source data 1.** *Figure 4* - source data for all protein intensity profiles.

**Figure supplement 1.** Analysis workflow for the 2D localization of proteins relative to actin cores with RIM technique.

**Figure supplement 2.** Localization in the sealing zone of cortactin, α-actinin 1, filamin A and vinculin.

**Figure supplement 3.** Localization in the sealing zone of paxillin and talin.

---

paxillin and talin, relatively to the intensity of F-actin staining of each podosome core. These analyses revealed a positive correlation between the actin content and each of the podosome components (*Figure 5A*). We then transduced human osteoclasts with mCherry-tagged LifeAct and GFP-tagged paxillin using two lentivirus constructs. Time-lapse RIM imaging of both F-actin and paxillin further confirmed that paxillin intensity fluctuations are locally correlated with F-actin intensity (*Figure 5B–C*, *Video 4*). These results suggested that, in addition to the local core coordination, there was also a local regulation between polymerization of actin in the core and accumulation of adhesion complex proteins.

Finally, we explored whether actin cores observed in the same cluster surrounded by adhesion complexes were synchronous. For this purpose, we analyzed how F-actin content within cores fluctuated relatively to each other inside the same clusters. The superposition of the fluorescence image and the segmentation of clusters of synchronized areas revealed that there was a synchrony within zones corresponding to multiple cores (*Figure 5D–F*, *Video 5*).

Altogether, these data demonstrated that the actin cores which were localized within the same islets were synchronized with each other and with the surrounding adhesion complexes.

## Discussion

This study provides a nanoscale picture of the architecture, spatial organization and dynamics of podosomes in the sealing zone of human osteoclasts adhering on bone. First, we gave a detailed insight into the inner organization of the sealing zone with the 3D localization of major actin cross-linkers inside the podosome belt, and examination of their axial localization hinted at the possible existence of tension within the integrin adhesion sites. Second, using RIM, we showed that podosome cores were within the submicron range from their direct neighbors. Third, by monitoring podosomes in the sealing zone by time-lapse super-resolution microscopy, we characterized the long-term spatial stability of actin cores during bone resorption and we revealed a synchronous behavior for actin cores within a distance of 1 µm. These fluctuations are probably the result of the constant renewal of the actin network in the sealing zone (*Saltel et al., 2004*), with a likely treadmilling of actin filaments from the plasma membrane towards the apical part of the cell. The Arp2/3 complex and cortactin are central in these dynamics (*Hurst et al., 2003*), and play a key role in the function of the sealing zone, probably via the generation of a protrusion force applied to the bone substrate, similarly to what was shown for macrophage podosomes (*Bouissou et al., 2017*; *Proag et al., 2015*; *Labernadie et al., 2014*). Finally, we found that synchronous neighbors were grouped within clusters, which corresponded to islets surrounded by adhesion complexes composed of vinculin, talin, paxillin and filamin A. Overall, these results allow for a new model of the internal architecture, dynamics and functioning of the sealing zone (*Figure 6*).

The evaluation of the 3D distributions of vinculin, paxillin, talin, filamin A, cortactin and α-actinin1 was carried out in the podosome belts of human osteoclasts on glass. Acquisitions were performed with the DONALD imaging technique, which combines dSTORM and SAF analysis for the efficient 3D detection of single fluorophores with a precision of 15 nm (*Bourg et al., 2015*). This 3D nanoscopy technique was recently applied in the context of human macrophage podosomes. It allowed for the identification of a close relationship between paxillin, vinculin and talin, and its requirement for

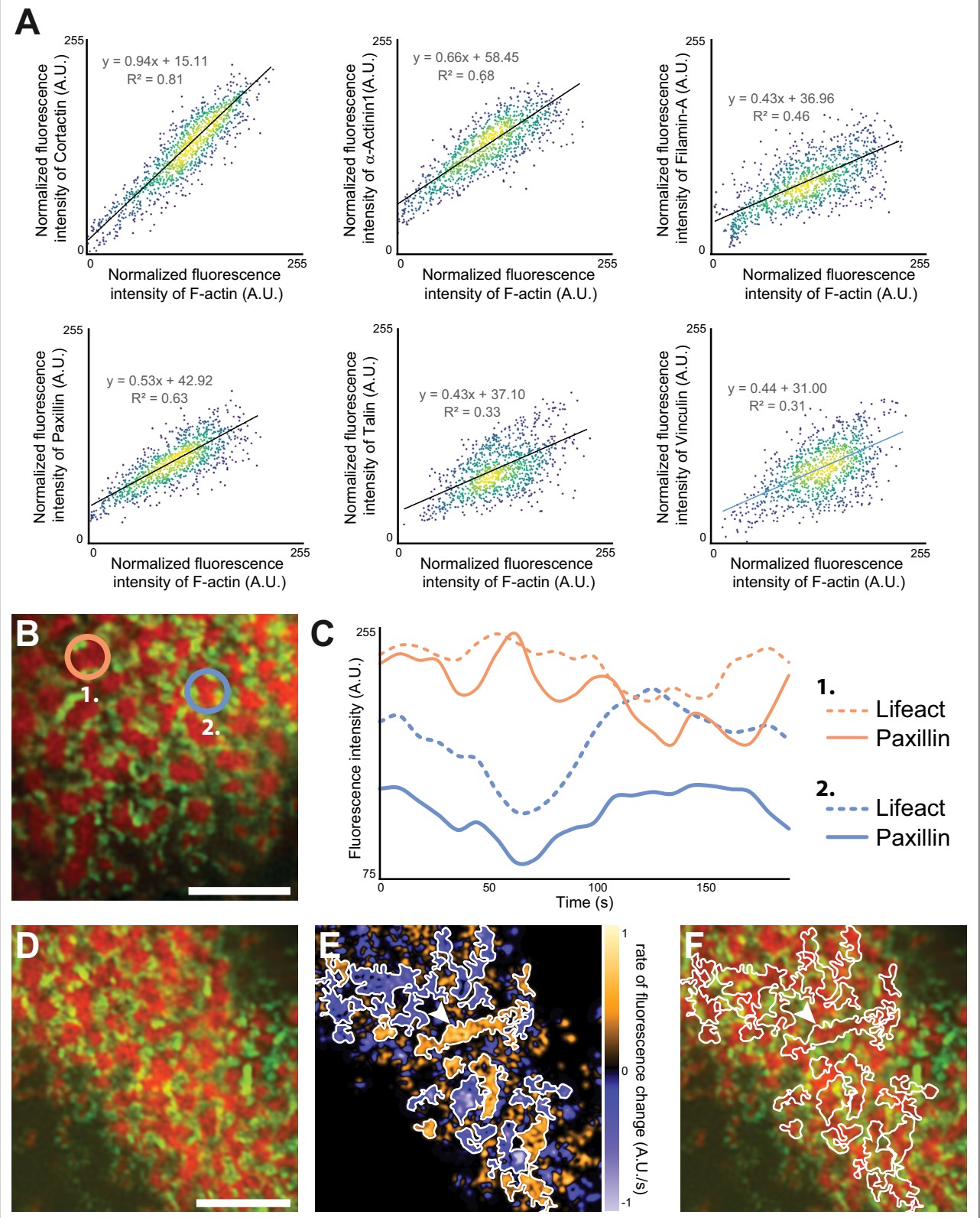

**Figure 5.** Quantification of the dynamics of the sealing zone. (**A**) Osteoclasts adhering to bone were stained for both F-actin and cortactin, α-actinin1, filamin A, vinculin, talin, or vinculin, respectively. The intensity of each fluorescent marker in 1 μm radius circles around F-actin cores were quantified for at least 1000 cores (in five cells from different donors), and correlated to the fluorescence intensity of F-actin. Data were normalized with respect to the maximum intensity. (**B–C**) Time-lapse RIM imaging of F-actin and paxillin in a living osteoclast expressing LifeAct-mCh and paxillin-GFP and adhering

*Figure 5 continued on next page*

Figure 5 continued

to bone. (**C**) The intensity variations of LifeAct-mCh and paxillin-GFP from two cores marked in (**B**) are shown. Note that the variations of the two podosome markers are correlated locally, but that the two cores, which are 5 µm apart, are not synchronized (**C**). (**D**) Time-lapse RIM imaging of F-actin and paxillin in a living osteoclast expressing LifeAct-mCh and paxillin-GFP and adhering to bone. A single RIM image of paxillin was acquired, followed by a stream acquisition of LifeAct-GFP, for a higher temporal resolution. (**E**) Image of the rate of fluorescence change corresponding to the cell shown in (**D**). (**F**) The superposition of the fluorescence image and the segmentation of clusters of synchronized areas in the sealing zone shows that there is a synchrony within zones corresponding to multiple cores encircled by paxillin. The arrowheads in (**E–F**) indicate a large cluster of actin cores. Note that all the different cores in this group are synchronized (in orange, **E**). Scale bars: 5 µm.

The online version of this article includes the following source data for figure 5:

**Source data 1.** *Figure 5* - source data for intensity correlation analysis between actin and all adhesion proteins.

**Source data 2.** *Figure 5* - source data for LifeAct-mCh and paxillin-GFP intensity variations for 2 cores.

efficient protrusion force generation (*Bouissou et al., 2017*). In osteoclast podosome belts, vinculin, paxillin, and talin-C were localized in the close vicinity of the ventral membrane. Moreover, vinculin and talin-C appeared to rise when situated closer to the actin cores. This interesting finding could be the first insight into the identification of possible tension within the podosome belt. In fact, in podosome rings of macrophages, it was observed that when talin is stretched, vinculin height is higher, probably because vinculin binding sites are appearing when talin acquires an extended conformation (*Bouissou et al., 2017*). In addition, cortactin, α-actinin1 and filamin A were mainly localized around 150 nm above the plasma membrane, with a dislocated distribution compared to actin. This suggests the existence of different sets of actin filaments within the sealing zone, those enriched in cross-linkers in an upper layer and those devoid of cross-linkers near the plasma membrane. Cryo-electron tomography analysis of the actin network composing the macrophage podosome suggests not only the central role of the actin core but also the concerted role of radial actin filaments (*Jasnin et al., 2021*). The present manuscript also suggests that, within podosome islets, different populations of actin filaments (actin cores positive for cortactin and lateral filaments positive for actinin or filamin-A) coexist (*Figure 6*) and are probably mechanically coupled and coordinated to allow force generation and efficient sealing of the bone breakdown zone.

Importantly, our results with this 3D super-resolution technique were obtained for podosomes belts of osteoclasts plated on glass. While only structures formed on bone or bone-mimicking substrate are called sealing zones (*Jurdic et al., 2006*; *Saltel et al., 2004*), podosome belts share some characteristics with sealing zones such as the local fusion of podosome cores. The use of RIM allowed for the identification of single actin cores within the sealing zones of human osteoclasts adhering on bone and resorbing. Podosome cores were within the submicron range from their direct neighbors, as already assessed (*Deguchi et al., 2019*). Additionally, we showed that the sealing zone was composed of sub-structures resembling islets of grouped podosomes within clusters and bordered by a network of adhesion sites. These observations greatly contrasted with the 'double circle' distribution (*i.e.* a dense network of actin cores that appear fused surrounded by two lines of adhesion sites) described in earlier works using conventional or

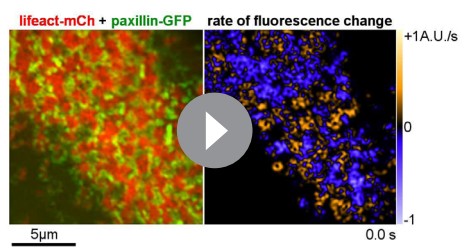

**Video 5.** RIM time-series and rate of fluorescence change of the F-actin content of a sealing zone, relative to the location of paxillin. Left panel: time-series movie by RIM microscopy of an osteoclast on bone expressing LifeAct-mCh and paxillin-GFP. A single RIM image of paxillin-GFP at the starting point of the movie was reconstructed and superimposed on a time series of LifeAct-GFP reconstructed at 2.4 s intervals during 88 s. Right panel: images of the rate of fluorescence change of LifeAct-GFP.
https://elifesciences.org/articles/75610/figures#video5

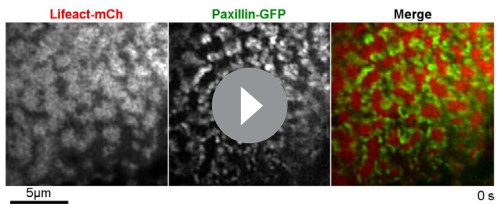

**Video 4.** Dynamics of F-actin and paxillin in a sealing zone. Time-series movie by RIM microscopy of an osteoclast on bone expressing LifeAct-mCh and paxillin-GFP, reconstructed at 9 s intervals during 160 s.
https://elifesciences.org/articles/75610/figures#video4

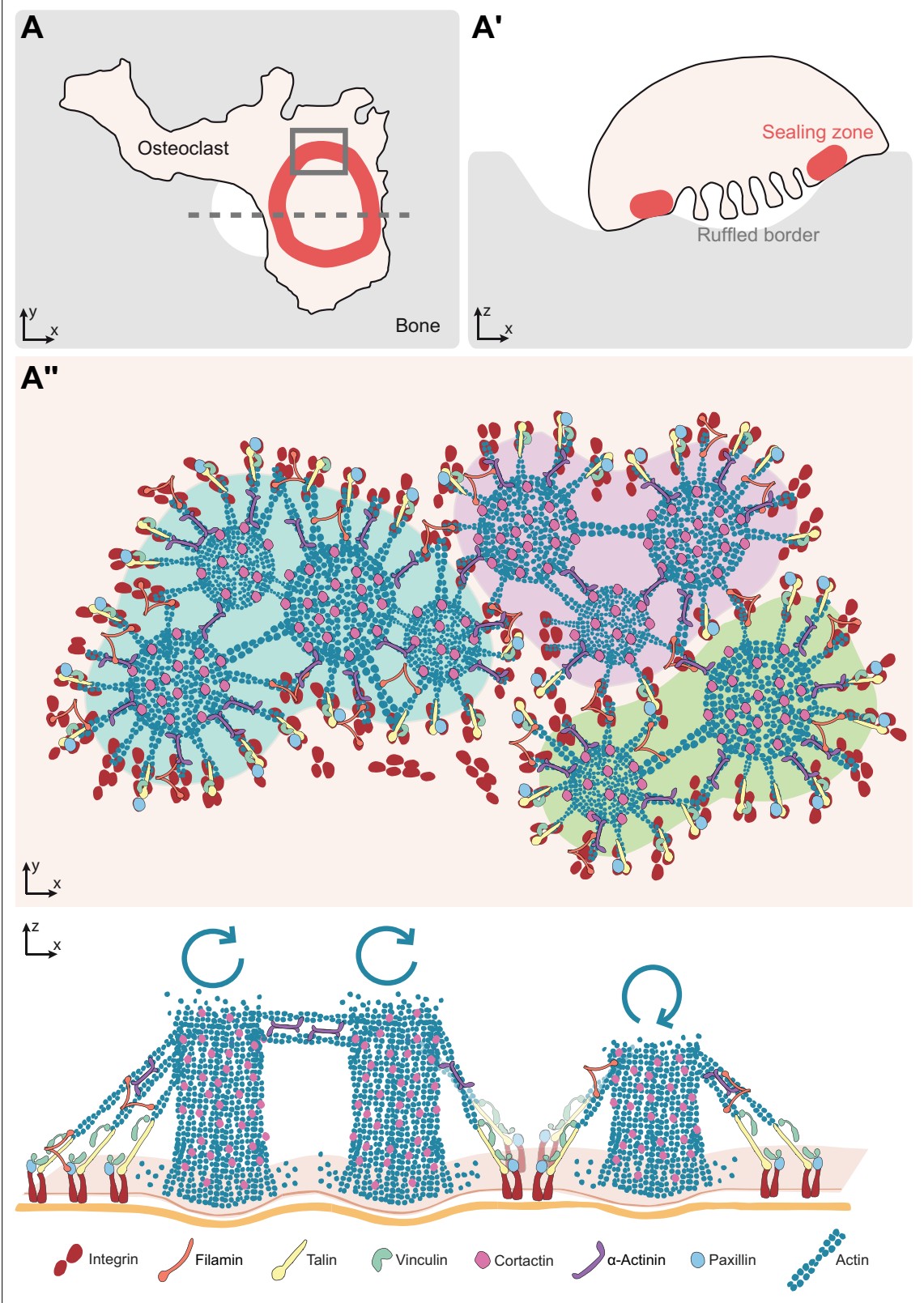

**Figure 6.** Model of the organization of the sealing zone into islets. (**A-A'**) Osteoclast form an actin-rich superstructure called the sealing zone, in order to confine bone degradation. (**A''**) Enlarged view of (**A**) to reveal the 3D organization into islets of coordinated actin cores, represented using different colors (upper panel). Actin cores that are localized within the same cluster tend to display synchronized actin fluctuations, which result from the respective rates of actin polymerisation at the plasma membrane and depolymerisation at the top of the structure, whereas there does not seem to be much correlation between clusters (lower panel). See legends for the different proteins.

confocal microscopy (*Lakkakorpi et al., 1993*; *Lakkakorpi and Väänänen, 1996*). This difference probably stems not only from the better resolution of the images presented in this manuscript but also from the fact that most of the studies on the organization of sealing zones were performed on mouse osteoclasts (*Jurdic et al., 2006*; *Saltel et al., 2008*, *Lakkakorpi et al., 2001*, *Ma et al., 2010*; *Lakkakorpi et al., 1999*; *Lakkakorpi and Väänänen, 1996*), and not on human osteoclasts.

Benefitting from RIM low toxicity and high temporal resolution, we could picture the sealing zone using time-lapse super-resolution microscopy. The actin dynamics was monitored within the sealing zone of osteoclasts plated on bone slices. Preliminary deconvoluted movies had revealed the apparent spatial stability of actin cores for long time periods. Actin signal processing yielded a time periodicity for these oscillations of approximately 100 s, which is much slower than the actin oscillations observed in scattered macrophage podosomes (*Bouissou et al., 2017*). Here, local oscillations associated with actin remodeling within the cores were revealed and characterized for the first time in the sealing zone. So far, the mechanisms regulating these oscillations, whether purely mechanical or involving signaling, as well as their importance for podosome and sealing zone function, are not yet understood. It was previously shown that these oscillations in macrophage podosomes depend on myosin IIA activity (*Labernadie et al., 2014*; *van den Dries et al., 2013*). It would thus be interesting to explore the effects of drugs interfering with actin polymerization on both the periodicity and the spatial synchrony properties of the sealing zone. Furthermore, cross-correlation analysis between signals associated with two cores in the same cell brought to light the existence of a spatial synchrony between neighbors up to an approximate 2 μm distance scale. Interestingly, this distance coincides with the mean neighbor-to-neighbor distance measured by SEM and RIM. Similar observations in human macrophage podosomes were reported, showing that under a distance of 1.8 μm, podosome neighbors oscillated synchronously (*Proag et al., 2015*). A role of connecting actin filaments was suggested to be involved in controlling the synchrony between podosomes (*Proag et al., 2015*; *Labernadie et al., 2014*). In sealing zones, the spectacularly dense actin meshwork observed between actin cores could be involved in connecting subunits and thus conducting synchronized oscillations of actin within the superstructure. In addition, our previous observations in macrophages revealed that actin oscillations corresponded to oscillations of the protrusion force generated by podosomes (*Proag et al., 2015*). Based on these observations in macrophages, we propose that in osteoclasts, synchrony of actin oscillations in actin core islets could reflect the involvement of oscillating protrusion forces in the sealing process. Actually, we also revealed the existence of intertwined islets of podosomes: some could be involved in sealing, while neighboring islets are concomitantly relaxing. A local co-regulation between actin polymerization in the core and formation of surrounding adhesion proteins was observed, as described in scattered podosomes of macrophages or dendritic cells (*Bouissou et al., 2017*; *van den Dries et al., 2013*). Thus, intermittent protrusive and relaxing capacity of podosome islets could enable efficient sealing of the osteoclast plasma membrane to the bone surface and maintaining the resorption lacuna and the diffusion barrier.

This study consists in the first extensive and quantitative study of the nanoscale organization and dynamics of the sealing zone in human osteoclasts. It proposes precise localization of six major components of the sealing zone: vinculin, paxillin, talin, cortactin, filamin A and α-actinin1. Furthermore, it provides the geometric characterization of the actin core network within this unique and functional superstructure. It also allows identification of dynamic processes related to podosomes during active bone resorption. This work therefore aimed at paving the way for future studies to decipher both the ultrastructural and dynamic properties of this unique osteoclast structure dedicated to bone resorption.

## Materials and methods
### Differentiation and culture of primary monocyte-derived osteoclasts
Human monocytes were isolated from blood of healthy donors as previously described (*Van Goethem et al., 2010*). Cells were re-suspended in cold PBS supplemented with 2 mM EDTA, 0.5% heat-inactivated fetal calf serum (FCS) at pH 7.4 and magnetically sorted with using magnetic microbeads coated with antibodies directed against CD14 (Miltenyi Biotec). Monocytes were then seeded on plastic at $2 \times 10^6$ cells/well in six-well plates in RPMI 1640 (Invitrogen) without FCS. After 2 h at 37 °C in humidified 5% $CO_2$ atmosphere, the medium was replaced by RPMI containing 10% FCS, 100 ng/

mL of macrophage colony-stimulating factor (M-CSF, Peprotech) and 60 ng/mL of human receptor activator of NF-κB-ligand (RANK-L, Miltenyi Biotec). The medium was then changed every third day by RPMI containing 10% FCS, 100 ng/mL of RANK-L and 25 ng/mL of M-CSF. For experiments, cells were harvested at day 10 using Accutase solution (Sigma-Aldrich) and centrifugation (1100 rpm, 10 min), and were then plated either on clean 1.5 H precision glass coverslips (Marienfeld 0117640) or on bovine bone slices (Immuno Diagnostic Systems DT-1BON1000-96). Cells were left to adhere in cytokine-supplemented medium for 3 days before fixation.

## Cleaning of precision glass coverslips

Before letting cells adhere on them, 1.5 H precision glass coverslips (Marienfeld) were cleaned as follows: they were placed on staining racks (Thermo Scientific 12627706) and immersed in a RBS 35 solution (Carl Roth 9238, 1/500 diluted in milliQ water) heated up to 80 °C while stirred with a magnetic bar. After 15 min, coverslips were rinsed three times with milliQ water and patted dry. They were put in a dry oven set at 175 °C for 120 min to sterilize. These clean coverslips were used within two weeks after RBS treatment.

## Primary antibodies

The following antibodies were used: goat anti-talin C-20 (Santa Cruz sc-7534, IF 1/50), mouse anti-vinculin clone hvin-1 (Sigma-Aldrich V9131, IF 1/50), mouse anti-paxillin (BD Biosciences 61005, IF 1/50), mouse anti-cortactin clone 4F11 (p80/85) (Sigma-Aldrich 05–180, IF 1/50), mouse anti-filamin A clone PM6/317 (Sigma-Aldrich MAB1678, IF 1/50), mouse anti-α-actinin1 clone BM-75.2 (Sigma-Aldrich A5044, IF 1/50) and mouse monoclonal antibody against the human LAMP-1 (H4A3, Santa Cruz Biotechnology, IF 1/100).

## Immunofluorescence

Osteoclasts plated on glass coverslips and bone slices for 3 days were fixed for 10 min in a 3.7% (wt/vol) paraformaldehyde (Sigma Aldrich 158127) solution containing 0.25% glutaraldehyde (Electron Microscopy Sciences 16220) in Phosphate Buffer Saline (PBS) (Fisher Scientific) at room temperature. When indicated, before fixation, cells were mechanically unroofed at 37 °C using distilled water containing protease inhibitors (Roche) and 10 μg/mL phalloidin (Sigma-Aldrich P2141) at 37 °C: cells were left still in this solution for 10 s, then a flow was created by flushing a dozen times so that cell dorsal membranes were ripped off. After fixation, quenching of free aldehyde groups was performed by treatment with PBS/50 mM ammonium chloride and PBS/1 mg/mL sodium borohydride. Non-unroofed cells were permeabilized for 10 min with PBS/0.3% Triton and all cells were blocked with PBS/1% BSA for 30 min. Samples were incubated with the primary antibodies for 90 min and then during 60 min with fluorescent dye conjugated-phalloidin and secondary antibodies for F-actin and proteins, respectively. For LAMP-1 staining, cells were permeabilized after fixation with 0.3% Saponin for 5 min, and then incubated with antibodies in PBS, 1% BSA and 0.06% Saponin.

## DONALD 3D super-resolution imaging

Osteoclasts plated on cleaned glass coverslips of accurate thickness (0.170+/-0.005 mm, Marienfeld 0117640) were unroofed and fixed as described above. Vinculin, paxillin, talin, cortactin, filamin A or α-actinin1 were stained with the corresponding primary antibody and an Alexa Fluor 647-coupled secondary antibody (Molecular Probes A21237, 1/1000) for dSTORM and podosome cores were labelled with Alexa Fluor 488-phalloidin (Molecular Probes A12379, 1/500) for epifluorescence. All dSTORM experiments were performed with the Smart-kit buffer (Abbelight, France).

3D super-localization images were acquired using an inverted IX83 microscope (Olympus) combined with SAFe module (Abbelight, France), and a TIRF module (Abbelight, France). Samples were excited with 405 nm (200 mW, ERROL Laser), 488 nm (150 mW, ERROL Laser), and 640 nm (400 mW, ERROL Laser) lasers in a HILO illumination (Highly Inclined and Laminated Optical sheet), and controlled via NEO Software (Abbelight, France). GFP / Alexa 532/mCherry / Alexa 647 fluorescence filters (Semrock, LF-405/488/532/635-B-OFF) were used, and the objective was a 100 x/1.49 N.A oil immersion objective (Olympus). All images were acquired using a sCMOS ORCA FLASH4.0 v3 (100 fps, cable camera link, Hamamatsu) camera, split on two regions of 300 × 300 pixels area and positioned on the focal plane of the SAFe module (×2.7 magnification, optical pixel size of 108 nm). The two imaging

paths are calibrated in terms of transmission efficiency to define a permanent correction factor that compensates the imperfect beam splitter. Images were collected once the density of fluorescent dye was sufficient (typically, under 1 molecule.µm$^{-2}$). About 5000 frames were recorded to compute one image of protein, and 500,000 frames were acquired to obtain one image of actin. For all recorded images, the integration time was set to 50ms and the EMCCD gain to 150. Laser power was adapted depending on the fluorophore density.

## Analysis of DONALD 3D super-resolution images

The super-localization of molecules and drift correction were performed on raw images *via* NEO Software (Abbelight, France) to achieve a final pixel size of 15 nm. Images were grouped in batches accounting for 5% of the total frames, and were submitted to drift correction by comparison batch per batch and according to a sliding window. Eventually, z correction was performed thanks to a dedicated Python script, to account for the calibration value of the SAFe module at the given moment of the experiment.

The spatial organization of proteins within the sealing zone like structure was characterized following an adapted version of the algorithm described previously (*Bouissou et al., 2017*). Briefly, actin cores in-plane coordinates ($x_{core}$,$y_{core}$) were determined from actin epifluorescence images from the 488 nm channel after Gaussian-filtering. The angle between the podosome belt portion and the horizontal was measured by the user, and the side pointing towards the interior of the cell identified. Then, centered on each core, a rectangle bounding box of length 10 µm and width 200 nm was drawn and the 3D coordinates ($x_i$, $y_i$, $z_i$) of the molecules inside this box were converted to a local r-z space ($r_i$, $z_i$) with $r_i$ being the distance between a molecule and the center of the considered actin core. This operation was repeated twice for each core: once along the structure direction, and once in the transversal direction. This analysis was performed for all cores of a given cell.

To further the analysis of the spatial organization of molecules within the podosome belt, a dedicated Python script was written to extract statistical information. Each cross-section direction for each cell was treated independently, except for distributions along r-axis and along z-axis for which all molecules detected for all cells were considered. Points were sorted in classes of varying lengths: 100 nm long until 1 µm from the core, 500 nm long further, depending on their distance to the corresponding actin core and whether they were located towards the exterior or the internal part of the cell. The distances were counted as negative when the molecules were located toward the exterior part of the cell. For each r-class, median height of the distribution was estimated cell by cell, by pooling the data for all cores considered within this particular cell. Also, the symmetry between internal and external distribution of proteins in terms of quantity was assessed by comparing the total amount of molecules on the exterior side to the total amount of points on the interior side, normalized by the total amount of points detected for each cell considered (*Figure 1—figure supplement 1*).

## Scanning electron microscopy imaging

Osteoclasts plated on bone slices for 3 days were unroofed as described above and fixed for 10 min in a 0.2 M sodium cacodylate buffer (pH 7.4) containing 2% paraformaldehyde (Electron Microscopy Sciences 15710) and 2.5% glutaraldehyde (Electron Microscopy Sciences 16220), then washed with distilled water. The samples were then prepared for observation following the protocol: they were dehydrated through a graded series (25–100%) of ethanol, transferred in acetone and subjected to critical point drying with $CO_2$ in a Leica EM CPD300, then sputter-coated with 3 nm platinum with a Leica EM MED020 evaporator, and were examined and photographed with an FEI Quanta FEG250.

## Analysis of SEM images

Actin cores were manually encircled using ImageJ oval selection tool, then the area and its center coordinates of these selected regions were extracted. A dedicated Python script was written to extract statistical information. The inter-core distances were computed using Delaunay tessellation on each image: for a given set P of discrete points in a general position is a triangulation such that no point in P is inside the circumcircle of any triangle. Delaunay triangulations maximize the minimum angle of all the angles of the triangles in the triangulation, not the edge-length of the triangles. The SciPy spatial algorithm 'Delaunay' was used in order to create the interconnection matrix, then the distance between each pair of connected vertices was computed and stored in a matrix. In order to

avoid plausible errors, the edges of the Delaunay tessellated space were excluded of the distance computations. To identify them, the SciPy spatial algorithm 'ConvexHull' was applied to the coordinates, and whenever two points were identified as pertaining to the convex envelope, the distance between them was not computed. Finally, for each vertex the minimum distance to all neighbors was kept for the nearest neighbor analysis.

## RIM 2D super-resolution imaging

For live imaging, osteoclasts were transduced with GFP- or mCherry-tagged LifeAct (*Riedl et al., 2008*) and GFP-paxillin lentiviruses (BiVic facility, Toulouse, France) 3 days as previously described (*Proag et al., 2015*), before being harvested and plated on bovine bone slices, and were observed the day after being plated on bone slices. Bone slices were placed on a FluoroDish (WPI FD35-100) with cells facing down and immersed with RPMI without phenol red, supplemented with 10% FCS (Thermo Fisher 32404–014). During observations, samples were maintained at 37 °C in humidified 5% $CO_2$ atmosphere. Images were acquired every 12ms for a total of 211 s (streams) using an inverted microscope (TEi Nikon). A fiber laser combiner with 4 fast diode lasers (Oxxius) with respective wavelengths 405 nm (LBX-405–180-CSB,) 454 nm (LBX-445–100-CSB), 488 nm (LBX-488–200-CSB), and 561 nm (LMX-561L-200-COL) were used to excite fluorophores. A corrected fiber collimator (RGBV Fiber Collimators 60FC Sukhamburg) was used to produce collimated $TEM_{00}$ 2.2 mm diameter output beam for all wavelengths. The polarization beam was rotated with an angle of 5 degrees before hitting a X4 Beam Expander beam (GBE04-A) and produced 8.8 mm beam $TEM_{00}$ beam. A fast spatial light phase binary modulator (QXGA fourth dimensions) was conjugated to the image plane to make speckle random illumination. The objective lens used in experiments is a ×100 magnification with 1.49 numerical aperture (CFI SR APO 100XH ON 1.49 DT 0;12 NIKON). A band pass filter was used for Green Fluorescence Protein (GFP) emission (Semrock FF01-514/30-25). A motorized high-speed wheel filter was used to sequentially turn the two band pass filters in 30ms after each 200 speckle fames. A piezoelectric Z stage (Z INZERT PIEZOCONCEPT) was used for fast z stack acquisition. For triggering, the camera sCMOS was used as master, and a rolling shutter output was used to trigger binary phase sequence to the SLM. The SLM output triggered the laser when binary phase mask was stable. A script from micromanager software was written to select the number of speckles used, the temporal resolution for each frame, the depth of z stack, the step of each z stack and the number of colors used for the acquisition. For stream recording, speckle frames were acquired continuously over the whole duration of the movie. Widefield time-lapse acquisitions were also carried out with the same microscope, but using a 60 x objective (CFI PLAN APO LBDA 60XH 1.4/0.13 NIKON). Images were captured every 2 s, and were restored with Huygens Software (classical maximum likelihood estimation with 30 iterations and theoretical PSF). For imaging of fixed samples, osteoclasts were unroofed and fixed as described above and stained for vinculin, cortactin, filamin A, or α-actinin1 with the corresponding primary antibody and an Alexa Fluor 488-coupled secondary antibody (Cell Signaling Technology #4408, 1/500) or an Alexa Fluor 546-coupled secondary antibody (Molecular Probes A11056, 1/500). Actin cores were labelled with Texas Red-phalloidin (Molecular Probes T7471, 1/200) or Alexa Fluor 488-phalloidin (Molecular Probes A12379, 1/200) respectively. Bone slices were placed in a FluoroDish, upside down on a droplet of Vectashield mounting medium (Vector Laboratories H-1000). Samples were excited with 488 nm and 561 nm laser diodes with the same setup as for time-lapse imaging. 200 speckle images for each channel were acquired sequentially to yield z stacks of various depths.

## Analysis of RIM images of fixed samples

Reconstruction of raw images was carried out as described in *Mangeat et al., 2021*. Briefly, the method is based on decreasing the computational cost of the inversion method described in *Idier et al., 2018*. This new method uses a variance matching process, instead of the marginal minimization methods based on the full covariance matrix of the data (*Mangeat et al., 2021*). The only input for the super-resolution reconstruction process is the knowledge of spatial statistics of speckle patterns limited by the OTF of the imaging system (Fourier transform of the PSF). Contrary to SIM, the exact knowledge of the illumination function is not necessary, and the protocol of the reconstruction is therefore drastically reduced. The inversion code is implemented on Matlab. The input are the excitation and collection PSF, generated with Gibson and Lanny 3D optical model implemented in the

plugin PSF generator (*Kirshner et al., 2013*). The PSF dimension is equal to the final size reconstruction with a pixel size equal to 32.25 nm for ×100 magnification. The position of the fluorophores is defined from the cover slide in each sample. The number of iterations during the variance matching process is defined by the user, mainly depending on the signal to noise ratio of the raw data.

Drift correction was performed on z stacks with ImageJ plugin Linear Stack Alignment with SIFT.

To create images on which to perform further analyses, 3–5 z slices per acquisition stack were selected based on their sharpness, depending on the quality of the original signal, and summed with ImageJ Z Project tool.

In order to characterize the actin network, actin cores were detected as local maxima using ImageJ Find Maxima tool with the threshold set as half of background intensity. The coordinates of these maxima were weighted considering all pixel intensity in a 200 nm radius, and exported in a text file. These points were positioned both on RIM reconstructed image and on its spatial derivative version created thanks to ImageJ Find Edges tool. From these locations, 8 radius profiles were traced with length 1 µm and width 100 nm on both images, and intensity values along each line were extracted and stored in a text file for each image. A dedicated Python script was then written to extract statistical data. Core radii were computed by detecting the first intensity maximum along the Find Edges profile. Inter-core distances were computed thanks to weighted coordinates following the same Delaunay algorithm already described in the SEM analysis section (*Figure 2—figure supplement 3*).

For the analysis of two-color acquisitions, actin cores were detected following the same procedure as described for the analysis of the actin network. Then, signals were extracted along 1.5 µm long and 100 nm wide lines drawn so that each core coordinates were placed at the middle of the line, and their orientation either followed the local curvature of the sealing zone, or was perpendicular to it. This process was repeated on the reconstructed actin image, the actin image after spatial derivation with ImageJ Find Edges tool and the protein image. All data were extracted and stored in a text file to be read by a dedicated Python script. This script computed the median core size for all data thanks to Find Edges signals, in order to establish a normalized profile length. This step was aimed at gaining independence from the specific geometric characteristics of each core, in order to propose a normalized profile for each signal and thus increasing the relevance of comparing data between cells from different blood donors. As a consequence, the x-axis for data displayed as signal profiles is shown as a non-unit scale based on the computed median core radius. Then, both actin- and protein-associated signals were interpolated along this new axis to yield comparable intensity profiles. Median profiles were eventually computed (*Figure 4—figure supplement 1*).

## Analysis of RIM live acquisitions

Reconstruction of raw images was carried out as described previously. To combine robust statistical estimation of object and temporal resolution, an interleaved reconstruction method was made as previously proposed for SIM (*Ma et al., 2018*). A total of 800 speckles were grouped to reconstruct one time slice, and the time step between two images corresponds to 200 speckles. Drift correction was performed on stacks with ImageJ plugin Linear Stack Alignment with SIFT, with the same parameters as for fixed samples. Actin intensity levels were normalized throughout the stack by using ImageJ Bleach Correction tool, with the correction method set to histogram matching.

To detect the single actin cores, all the time slices were summed with ImageJ Z project tool and the coordinates were extracted according to the same procedure as for fixed samples. The dynamic characteristics of actin were assessed by extracting the time-dependent signals in a circular selection of radius 100 nm around each core, and storing them in a text file. A dedicated Python script was developed to compute the distance between all core pairs in the same cell, the Fourier spectrum associated with each signal and the Pearson cross-correlation between two signals in the same cell. Natural frequencies were identified as the frequencies associated with Fourier coefficients greater than a threshold value proportional to the median value for Fourier coefficients over the spectrum. Pearson coefficients were eventually sorted according to the distance between the core pair coordinates.

In order to obtain graphical representations of the local evolution of fluorescence over time, as shown in *Figures 3 and 5* and *Videos 3 and 5*, the films were processed using ImageJ as follows. First, the films were registered using the StackReg ImageJ plugin. Then, 2 sequential time points were subtracted to obtain a new film representing the rate of change of fluorescence. Finally, a Gaussian

filter (2 pixel radius, *i.e.* 64.5 nm) and a time average on three consecutive images were applied to this film to help reduce local noise and visually highlight local variations in intensity changes.

## Statistical analysis

All box-and-whisker plots show the median, lower and upper quartiles (box) and the 10th and 90th percentiles (whiskers).

## Resource availability

All data generated or analysed during this study are included in the manuscript and supporting files; Source Data files have been provided for *Figures 1–5*.

## Acknowledgements

The authors are grateful to Myriam Ben Neji for isolation of human blood monocytes and Isabelle Fourquaux from TRI imaging facility for SEM preparation. The authors also acknowledge Anne Blangy, Alessandra Cambi, Amsha Proag and Olivier Destaing for helpful discussions. This work has been supported in part by l'Agence Nationale de la Recherche (ANR16-CE13-MechanOCs), l'Université de Toulouse, la Région Occitanie, la Fondation pour la Recherche Médicale (FRM DEQ2016 0334894), INSERM Plan Cancer and Human Frontier Science Program (RGP0035/2016).

## Additional information

### Funding

| Funder | Grant reference number | Author |
|---|---|---|
| Agence Nationale de la Recherche | ANR16-CE13 | Christel Vérollet |
| Fondation pour la Recherche Médicale | FRM DEQ2016 0334894 | Isabelle Maridonneau-Parini |
| Human Frontier Science Program | RGP0035/2016 | Isabelle Maridonneau-Parini |

The funders had no role in study design, data collection and interpretation, or the decision to submit the work for publication.

### Author contributions

Marion Portes, Conceptualization, Data curation, Formal analysis, Funding acquisition, Investigation, Methodology, Project administration, Resources, Software, Validation, Visualization, Writing - original draft, Writing – review and editing; Thomas Mangeat, Conceptualization, Data curation, Formal analysis, Funding acquisition, Investigation, Methodology, Project administration, Software, Supervision, Validation, Visualization, Writing - original draft, Writing – review and editing; Natacha Escallier, Data curation, Formal analysis, Investigation, Methodology, Resources, Software, Visualization, Writing – review and editing; Ophélie Dufrancais, Conceptualization, Investigation, Methodology, Software, Validation, Visualization, Writing – review and editing; Brigitte Raynaud-Messina, Conceptualization, Formal analysis, Funding acquisition, Investigation, Methodology, Software, Supervision, Validation, Visualization, Writing – review and editing; Christophe Thibault, Conceptualization, Formal analysis, Funding acquisition, Investigation, Methodology, Project administration, Resources, Software, Supervision, Validation, Writing – review and editing; Isabelle Maridonneau-Parini, Conceptualization, Funding acquisition, Investigation, Project administration, Resources, Supervision, Validation, Writing - original draft, Writing – review and editing; Christel Vérollet, Conceptualization, Data curation, Formal analysis, Funding acquisition, Investigation, Project administration, Resources, Software, Supervision, Validation, Visualization, Writing - original draft, Writing – review and editing; Renaud Poincloux, Conceptualization, Data curation, Formal analysis, Funding acquisition, Investigation, Methodology, Project administration, Resources, Software, Supervision, Validation, Visualization, Writing - original draft, Writing – review and editing

### Author ORCIDs
Marion Portes (iD) http://orcid.org/0000-0002-3539-2425
Brigitte Raynaud-Messina (iD) http://orcid.org/0000-0002-7637-1997
Christel Vérollet (iD) http://orcid.org/0000-0002-1079-9085
Renaud Poincloux (iD) http://orcid.org/0000-0003-2884-1744

### Ethics

Monocytes from healthy subjects were provided by Etablissement Français du Sang (EFS), Toulouse, France, under contract 21/PLER/TOU/IPBS01/20130042. According to articles L12434 and R124361 of the French Public Health Code, the contract was approved by the French Ministry of Science and Technology (agreement number AC 2009921). Written informed consents were obtained from the donors before sample collection.

### Decision letter and Author response
Decision letter https://doi.org/10.7554/eLife.75610.sa1
Author response https://doi.org/10.7554/eLife.75610.sa2

---

## Additional files

### Supplementary files
• Transparent reporting form

### Data availability
All data generated or analysed during this study are included in the manuscript and supporting files; Source Data files have been provided for Figures 1, 2, 3, 4 and 5.

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
