## [Editor Report]

The authors present here an elegant study in which they analyze the structure of human osteoclast podosomes at the nanoscale, combining microscopy methods to reveal the architecture and dynamics of the sealing zone on the bone surface. Through random illumination microscopy, the live imaging of densely packed actin cores within the sealing zone shows that these cores are locally synchronized, connected by α-actinin filaments, and surrounded by adhesion complexes. The authors propose a model in which the function of podosomes during bone resorption is accomplished through the coordination of islets of the actin core and not through the global coordination of all podosome subunits that form the sealing zone. This article has the potential to generate a significant impact in the field of osteoclast biology.

---

## [Decision Letter]

**Decision letter after peer review:**

Thank you for submitting your article "Nanoscale architecture and coordination of actin cores within the sealing zone of human osteoclasts" for consideration by *eLife*. Your article has been reviewed by 2 peer reviewers, and the evaluation has been overseen by a Reviewing Editor and Mone Zaidi as the Senior Editor. The following individual involved in review of your submission has agreed to reveal their identity: L. Shannon Holliday (Reviewer #1).

Essential revisions:

Reviewers noted the importance of this work whose major strengths are the state-of-the-art imaging approach, the use of quantitative analysis, and the importance of a study on primary osteoclasts cultured on bone slices, given that podosome conformations are highly dependent on the type of substrate that should accurately mimic the physiological environment in vivo. However, some issues were identified that may require further experimental support and/or explicit discussion in the text.

1. Provide a better description to support the hypothesis that actin flows through filaments by polymerization and depolymerization. In particular, improve the summary Figure 6 showing synchronized actin fluctuations that appear to reflect the persistence of the cores, not necessarily the rate of actin polymerization and depolymerization.

2. To improve novel aspects of the results obtained, the authors should measure parameters such as group stability, number of podosomes per group, and group size.

*Reviewer #1 (Recommendations for the authors):*

The manuscript should serve as a valuable contribution to the literature but I had a number of concerns that should be addressed prior to publication.

1. Introduction. In the manuscript the author's fail to cite King, GJ and Holtrop ME.1975. Actin-like filaments in bone cells of cultured mouse calvaria as demonstrated by binding to heavy meromyosin. J. Cell Biol. 66(2):445-451. Because this manuscript was literally more than a decade ahead of its time it is often missed, but using traditional TEM provides a very nice picture of the actin rings of resorbing osteoclasts. It would be interesting perhaps to compare the models of the structures established in this study to the structures described in this manuscript. Pretty similar I think.

2. A key point that has been made about actin rings is the extreme dynamics of driven by polymerization of the filaments near the sealing zone membrane and depolymerization toward the middle of the cells. This was described elegantly by Saltel and colleagues (ref 23). In the current manuscript the persistence of podosomes in examined but the idea that actin is flowing through the filaments by polymerization and depolymerization, and is probably pushing the network into the membrane is not well described, and provides important context for the study. For example, the authors note that cortactin is in the core of the podosomes as opposed to other proteins. A description of why, because cortactin is associated with the n-WASP/ARP2/3 complex that regulates actin polymerization would help to bring the story together.

3. Figure 2. (A) The pit formed by the osteoclast is not very convincing. It would be more convincing to show bone slices on which the cells were removed were pitted extensively. Since this part of the study involves resorbing osteoclasts, it would also be helpful if the ruffled membrane was shown, using appropriate antibody markers.

B) It was not clear to me what I am looking at, it looks like that if it is an actin ring there is very little ruffled membrane area. This happens as resorption complexes form or end their time. Maybe a little schematic?

C) I was not entirely clear how the heights being measured were established. Better description would be helpful. Maybe it is present in the Methods but it is not presented very clearly.

4. Please provide in the manuscript a bit more detail regarding the actin and paxillin constructs that were transduced. Do we know that the actin and paxillin function normally in cells? If so, please state and cite appropriate sources. Also, how much of the exogenous protein is being expressed? Is it enough to cause changes in the normal protein balances and in cell behavior?

5. In Figure 6, which is a nice figure, but please explain in more detail what is being shown. For example, I think A" is the square from A but please be clear. The color coding I assume indicate islets within the sealing zone. The synchronized actin fluctuations seem to reflect the persistence of the cores, not necessarily the speed of actin polymerization and depolymerization.

What do the little circles with the arrows mean in the bottom figure? Different synchronization?

In the model, integrins and associated proteins are attached to the core podosome, and the podosomes are connected through non-podosomal microfilaments. This seems reasonable but it is not clear how these attachments occur since the individual monomers in the podosome cores are flowing through the podosomes, entering at the membrane and exiting when the filaments depolymerize. This would exert tension, but it seems these must be transient, or subsets of the filaments might get stuck under the traction and not display the dynamics of the overall podosomes. The study in reference 23 looked at osteoclasts on apatite, so specific integrin ligand interactions presumably were not occurring. It would be perhaps interesting to do similar studies to what you describe on bone, on hydroxyapatite, to see if there were changes in the filament dynamics if integrin ligands were not readily available (though not for this manuscript).

In general, I think this is an excellent manuscript and will be a valuable contribution to the literature. For me it was a little confusing in places because certain parameters were not well described, and in my view, the article would have greater impact if it was a bit more transparent. I think probably the descriptions in the methods section might be adequate for people who do these studies and use these techniques regularly, but leave the less well informed reader more puzzled than they need be.

It would be helpful also to compare the parameters of the podosomes and actin rings described here with those described earlier using other techniques.

*Reviewer #2 (Recommendations for the authors):*

The references to the Figures are not always correct and I would encourage the authors to carefully check these in the manuscript. I would also encourage the authors to reconsider their labeling since the usage of the many (sub)accents in the panel labeling is confusing. I would further encourage the authors to for example add a label that to the image panels that indicates what is visualized. Currently, you need to go to the Figure legends for this information which is quite inconvenient.

---

## [Author Response]

Essential revisions:Reviewers noted the importance of this work whose major strengths are the state-of-the-art imaging approach, the use of quantitative analysis, and the importance of a study on primary osteoclasts cultured on bone slices, given that podosome conformations are highly dependent on the type of substrate that should accurately mimic the physiological environment in vivo. However, some issues were identified that may require further experimental support and/or explicit discussion in the text.1. Provide a better description to support the hypothesis that actin flows through filaments by polymerization and depolymerization. In particular, improve the summary Figure 6 showing synchronized actin fluctuations that appear to reflect the persistence of the cores, not necessarily the rate of actin polymerization and depolymerization.

Indeed, the actin network in the sealing zone is in constant renewal (Saltel et al., Mol Biol Cell 2004), with a likely treadmilling of actin filaments from the plasma membrane towards the apical part of the cell. These dynamics probably play a key role in the function of the sealing zone, notably via the generation of a protrusion force applied to the bone substrate, similarly to what we have shown for macrophage podosomes (Labernadie et al., Nat. Commun. 2014; Proag et al., ACS Nano 2015; Bouissou et al., ACS Nano 2017). In this context, actin branching, mediated by cortactin and the Arp2/3 complex (Hurst et al., J Bone Miner Res 2004), most likely plays a central role in the ability of the sealing zone to generate protrusive forces and seal the bone degradation zone. We have improved the summary Figure 6 and now comment this at the beginning of the discussion, see P6, L245.

2. To improve novel aspects of the results obtained, the authors should measure parameters such as group stability, number of podosomes per group, and group size.

We have quantified the number of podosomes per group and the group size.

Measurements of these islets of clustered cores showed that they were 2.3 +/-2.1 µm² (average +/-SD) and contained in 7 +/-8 (average +/-SD) cores. These results are now included in the manuscript (P6 L213). Unfortunately, we could not accurately measure the stability of the clusters, as this would require a long, and challenging, time-lapse by RIM of osteoclasts expressing both paxillin-GFP and lifeact-mCherry, which we were able to achieve only on a few cells and on short timescales.

Reviewer #1 (Recommendations for the authors):The manuscript should serve as a valuable contribution to the literature but I had a number of concerns that should be addressed prior to publication.1. Introduction. In the manuscript the author's fail to cite King, GJ and Holtrop ME.1975. Actin-like filaments in bone cells of cultured mouse calvaria as demonstrated by binding to heavy meromyosin. J. Cell Biol. 66(2):445-451. Because this manuscript was literally more than a decade ahead of its time it is often missed, but using traditional TEM provides a very nice picture of the actin rings of resorbing osteoclasts. It would be interesting perhaps to compare the models of the structures established in this study to the structures described in this manuscript. Pretty similar I think.

We thank the reviewer for this suggestion. This reference is indeed too often omitted. We have corrected this error by now referring to it when presenting the sealing zone in the introduction of the manuscript (P2, L46).

2. A key point that has been made about actin rings is the extreme dynamics of driven by polymerization of the filaments near the sealing zone membrane and depolymerization toward the middle of the cells. This was described elegantly by Saltel and colleagues (ref 23). In the current manuscript the persistence of podosomes in examined but the idea that actin is flowing through the filaments by polymerization and depolymerization, and is probably pushing the network into the membrane is not well described, and provides important context for the study. For example, the authors note that cortactin is in the core of the podosomes as opposed to other proteins. A description of why, because cortactin is associated with the n-WASP/ARP2/3 complex that regulates actin polymerization would help to bring the story together.

Indeed, the actin network in the sealing zone is in constant renewal (Saltel et al., Mol Biol Cell 2004), with a likely treadmilling of actin filaments from the plasma membrane towards the apical part of the cell. These dynamics probably play a key role in the function of the sealing zone, notably via the generation of a protrusion force applied to the bone substrate, similarly to what we have shown for macrophage podosomes (Labernadie et al., Nat. Commun. 2014; Proag et al., ACS Nano 2015; Bouissou et al., ACS Nano 2017). In this context, actin branching, mediated by cortactin and the Arp2/3 complex (Hurst et al., J Bone Miner Res 2004), most likely plays a central role in the ability of the sealing zone to generate protrusive forces and seal the bone degradation zone. Our recent work on the structure of the actin network of the macrophage podosome (Jasnin et al., Nat. Commun. 2022, in press, available at BioRxiv) suggests not only the central role of the actin core but also the concerted role of radial actin filaments. The present manuscript also suggests that, within modules that we call podosome islets, different populations of actin filaments coexist: actin cores positive for cortactin and lateral filaments positive for actinin or filamin-A, and it seems logical that these different networks are mechanically coupled and coordinated to allow force generation and efficient sealing of the bone breakdown zone. We now discuss this P6, L245.

3. Figure 2. (A) The pit formed by the osteoclast is not very convincing. It would be more convincing to show bone slices on which the cells were removed were pitted extensively. Since this part of the study involves resorbing osteoclasts, it would also be helpful if the ruffled membrane was shown, using appropriate antibody markers.

In order to appreciate their bone degrading activity, we now show a gallery of scanning electron micrographs of human osteoclast on bone (new Figure 2-supplemental figure 1A), as well as, as suggested by the reviewer, images of degraded bone slices without osteoclasts (new Figure 2supplemental figure 1B). In addition, as requested by the reviewer, we made some experiments and now provide pictures of immunostainings of LAMP1 positive compartments that confirm the accumulation of secretary lysosomes in the middle of the sealing zone (new Figure 2supplemental figure 2C).

B) It was not clear to me what I am looking at, it looks like that if it is an actin ring there is very little ruffled membrane area. This happens as resorption complexes form or end their time. Maybe a little schematic?

Figure 2B shows the apical part of an osteoclast that was unroofed on bone. As noted by the reviewer, there is no ruffled membrane visible in the middle of the sealing zone. The reason is that the plasma membrane at the ruffled border could not be preserved by the unroofing procedure, since we can directly observe the bone under the cell, with it typical cracks. We now clarify that the ruffled membrane is not visible in the corresponding Figure legend.

C) I was not entirely clear how the heights being measured were established. Better description would be helpful. Maybe it is present in the Methods but it is not presented very clearly.

For Figure 2C and Video 1, a z-stack of an osteoclast stained for F-actin and adhering on bone was acquired every 200 nm and reconstructed by RIM. Each plane of this z-stack has been colored with a different color. Video 1 shows the entire z-stack while Figure 2C shows the z projection of the stack. This is now better explained in the corresponding Figure legend and Video 1 legend.

4. Please provide in the manuscript a bit more detail regarding the actin and paxillin constructs that were transduced. Do we know that the actin and paxillin function normally in cells? If so, please state and cite appropriate sources. Also, how much of the exogenous protein is being expressed? Is it enough to cause changes in the normal protein balances and in cell behavior?

It is difficult to evaluate how much these exogenous proteins are expressed as very few cells are actually expressing them, but the lifeact construct expressed to label actin filaments is widely used and do not affect actin dynamics (Riedl, J. et al., Nat Methods 2010). This reference has been added in the manuscript. We also observed osteoclasts expressing lifeact with or without paxillin-GFP and observed no effect of exogenous paxillin expression on the sealing zone dynamics.

5. In Figure 6, which is a nice figure, but please explain in more detail what is being shown. For example, I think A" is the square from A but please be clear.

We thank the reviewer for this comment. We now explicit in the Figure legend that A" is the square from A”.

The color coding I assume indicate islets within the sealing zone.

This is the case. This is now also specified in the Figure legend.

The synchronized actin fluctuations seem to reflect the persistence of the cores, not necessarily the speed of actin polymerization and depolymerization.What do the little circles with the arrows mean in the bottom figure? Different synchronization?

Indeed, the same circular arrows for the left-side cores represent their quasi-synchronous state, whereas the different circular arrow for the right-side core denotes its lack of synchronicity with the two others.

In the model, integrins and associated proteins are attached to the core podosome, and the podosomes are connected through non-podosomal microfilaments. This seems reasonable but it is not clear how these attachments occur since the individual monomers in the podosome cores are flowing through the podosomes, entering at the membrane and exiting when the filaments depolymerize. This would exert tension, but it seems these must be transient, or subsets of the filaments might get stuck under the traction and not display the dynamics of the overall podosomes. The study in reference 23 looked at osteoclasts on apatite, so specific integrin ligand interactions presumably were not occurring. It would be perhaps interesting to do similar studies to what you describe on bone, on hydroxyapatite, to see if there were changes in the filament dynamics if integrin ligands were not readily available (though not for this manuscript).

The dynamics of the individual components of the podosome and how it contributes to the dynamics and function of the sealing zone are indeed intriguing. So far, the respective functions of integrins and their interactors have been poorly studied in osteoclasts. Comparison of osteoclast adhesion on bone and hydroxyapatite would be informative, and we have also already begun to investigate the regulation of integrin activation in osteoclasts, but, as the reviewer conceded, that is beyond the scope of this article.

In general, I think this is an excellent manuscript and will be a valuable contribution to the literature. For me it was a little confusing in places because certain parameters were not well described, and in my view, the article would have greater impact if it was a bit more transparent. I think probably the descriptions in the methods section might be adequate for people who do these studies and use these techniques regularly, but leave the less well informed reader more puzzled than they need be.

In order to facilitate the understanding of the manuscript by the largest number of readers, we had it read by non-specialists, which allowed us to clarify and make explicit certain parts (in particular in the Materials and methods: changes are highlighted in yellow).

It would be helpful also to compare the parameters of the podosomes and actin rings described here with those described earlier using other techniques.

The parameters of the podosome cores (size and distance between cores, for instance) within the sealing zone were compared with previous studies performed in osteoclasts and macrophages. Strikingly, actin cores in osteoclasts (this manuscript) were smaller compared to those of macrophage podosomes (97-114 nm Figure 2D) vs 150-200 nm (Proag et al., ACS Nano 2015; Jasnin et al., Nat Commun 2022, in press). The inter-podosome distance in osteoclasts (705 nm for direct neighbors (Figure 2E) and 443 nm for first neighbors (Figure 2F)) was also reduced, compared to macrophage podosomes (1770 nm for direct neighbors, Proag et al., ACS Nano 2015) and similar to what was measured in osteoclasts by (Deguchi et al., J Phys D Appl Phys 2020). This is now described in the results P4, L130.

Reviewer #2 (Recommendations for the authors):The references to the Figures are not always correct and i would encourage the authors to carefully check these in the manuscript. I would also encourage the authors to reconsider their labeling since the usage of the many (sub)accents in the panel labeling is confusing. I would further encourage the authors to for example add a label that to the image panels that indicates what is visualized. Currently, you need to go to the Figure legends for this information which is quite inconvenient.

We carefully revised all the references to figures in the main text. As suggested, we also reduced the number of figure labels with (sub)accents to keep them only in the case of different magnification of the same image and labelled the panel with the name of the visualized proteins.